# Identification and outcomes of acute kidney disease in patients presenting in Bolivia, Brazil, South Africa, and Nepal

Rhys D. R. Evans[1]*, Sanjib K. Sharma[2], Rolando Claure-Del Granado[3,4], Brett Cullis[5], Emmanuel A. Burdmann[6], FOS Franca[6], Junio Aguiar[7], Martyn Fredlund[8], Kelly Hendricks[9], Maria F. Iturricha-Caceres[10], Mamit Rai[2], Bhupendra Shah[2], Shyam Kafle[2], David C. Harris[11], Mike V. Rocco[12]

1 Centre for Kidney and Bladder Health, University College London, London, United Kingdom, 2 B.P. Koirala Institute of Health Sciences, Dharan, Nepal, 3 IIBISMED, Universidad Mayor de San Simon, School of Medicine, Cochabamba, Bolivia, 4 Division of Nephrology, Hospital Obrero No 2 –CNS, Cochabamba, Bolivia, 5 University of Cape Town, Cape Town, South Africa, 6 LIM 12, Division of Nephrology, and Department of Infectious and Parasitic Diseases, University of Sao Paulo, Medical School, Sao Paulo, Brazil, 7 University of Para, Santarem, Brazil, 8 North Bristol NHS Trust, Bristol, United Kingdom, 9 ISN Programs, Denver, Colorado, United States of America, 10 Facultad de Medicina, Universidad Privada del Valle, Tiquipaya, Bolivia, 11 Westmead Institute for Medical Research, University of Sydney, Sydney, Australia, 12 Wake Forest School of Medicine, Winston-Salem, North Carolina, United States of America

* Rhys.evans5@nhs.net

## Abstract

### Background

The International Society of Nephrology proposes an acute kidney disease (AKD) management strategy that includes a risk score to aid AKD identification in low- and low-middle-income countries (LLMICs). We investigated the performance of the risk score and determined kidney and patient outcomes from AKD at multiple LLMIC sites.

### Methods and findings

Adult patients presenting to healthcare facilities in Bolivia, Brazil, South Africa, and Nepal were screened using a symptom-based risk score and clinical judgment. Those at AKD risk underwent serum creatinine testing, predominantly with a point-of-care (POC) device. Clinical data were collected prospectively between September 2018 and November 2020. We analyzed risk score performance and determined AKD outcomes at discharge and over follow-up of 90 days. A total of 4,311 patients were at increased risk of AKD, and 2,922 (67.8%) had AKD confirmed. AKD prevalence was 80.2% in patients enrolled based on the risk score and 32.5% when enrolled on clinical judgment alone ($p < 0.0001$). The area under the receiver operating characteristic curve was 0.73 for the risk score to detect AKD. Death during admission occurred in 84 (2.9%) patients with AKD and 3 (0.2%) patients without kidney disease ($p < 0.0001$). Death after discharge occurred in 206 (9.7%) AKD patients, and 1865 AKD patients underwent reassessment of kidney function after discharge; 902 (48.4%) patients had persistent kidney disease including 740 (39.7%) patients reclassified with de novo or previously undiagnosed chronic kidney disease (CKD). The study was

**Data Availability Statement:** Original data is owned by the International Society of Nephrology

and will be made freely available upon reasonable request (research@theisn.org).

**Funding:** The study was funded by a grant from the Stavros Niarchos Foundation (https://www.snf.org), which was provided directly to the International Society of Nephrology (through its executive director in 2017, Luca Segantini). Funds for the project were subsequently managed by KH on behalf of the ISN. The funders had no role in study design, data collection and analysis, decision to publish, or preparation of the manuscript.

**Competing interests:** RE has received honoraria from Therakos. EB has received honoraria from Baxter and AstraZeneca. Other authors have no conflicts to declare.

**Abbreviations:** ADQI, Acute Disease Quality Initiative; AKD, acute kidney disease; AKI, acute kidney injury; CI, confidence interval; CKD, chronic kidney disease; eGFR, estimated glomerular filtration rate; ESKD, end-stage kidney disease; HCC, healthcare center; IQR, interquartile range; ISN, International Society of Nephrology; KCN, Kidney Care Network; KRT, kidney replacement therapy; LLMIC, low- and low-middle-income country; NKD, no kidney disease; OR, odds ratio; POC, point-of-care; SD, standard deviation.

pragmatically designed to assess outcomes as part of routine healthcare, and there was heterogeneity in clinical practice and outcomes between sites, in addition to selection bias during cohort identification.

## Conclusions

The use of a risk score can aid AKD identification in LLMICs. High rates of persistent kidney disease and mortality after discharge highlight the importance of AKD follow-up in low-resource settings.

---

## Author summary

### Why was this study done?

- Acute kidney disease (AKD) is common in low-resource settings and leads to preventable deaths.
- Management strategies that may improve outcomes for patients with AKD are untested when used within routine clinical care in low- and low-middle-income countries (LLMICs).
- Outcomes after AKD episode in LLMICs are unknown.

### What did the researchers do and find?

- The researchers tested the performance of a scoring system to screen patients for risk of AKD in Bolivia, Brazil, Nepal, and South Africa and determined kidney and patient outcomes in the 3 months after AKD episode.
- The risk score was effective in screening patients for AKD; 48.2% of patients with AKD had persistent kidney disease after 3 months.

### What do these findings mean?

- Use of a risk score facilitates AKD identification and can be incorporated within routine clinical care in LLMICs.
- Frameworks need to be developed that allow patient follow-up after AKD episode as opportunities for chronic kidney disease (CKD) diagnosis are currently being missed.
- There were differences in the clinical approach taken at different sites and whether the management strategy improves patient outcomes in the longer term is unknown.

## Introduction

Kidney disease disproportionately affects disadvantaged populations in low- and low-middle-income countries (LLMICs) with poor access to care [1,2]. Greater than 850 million people are affected by kidney disease worldwide, with the majority of patients (e.g., 59% to 64% of chronic kidney disease [CKD]) concentrated in LLMICs [3–5]. Acute kidney disease (AKD) is common in these settings and is particularly important as it frequently affects young patients but often goes unrecognized or untreated leading to high mortality from the acute episode and may not recover leading to the development of CKD [6,7]. Undiagnosed and untreated CKD may progress to end-stage kidney disease (ESKD), which has devastating impacts for individuals and health systems in LLMICs; 90% of disadvantaged populations have no access to kidney replacement therapy (KRT) and when KRT can be provided it comes at significant economic cost. Moreover, there is an 80-fold difference in the number of nephrologists between low- and high-income countries, highlighting major deficiencies in LLMIC ability to provide good quality kidney care [2,5]. To address global inequities in kidney care, the International Society of Nephrology (ISN) launched the 0by25 initiative in 2013 with the ultimate aim of eliminating preventable deaths from acute kidney injury (AKI), present in a subset of patients with AKD, with a particular emphasis on people living in LLMICs [8,9]. An ambitious target was set to try and achieve this aim by 2025.

A key challenge in the management of AKD in LLMICs is the ability to identify patients at AKD risk and the capability to confirm AKD diagnosis with prompt serum creatinine (SCr) testing. This challenge is a consequence, at least in part, of a deficiency in nephrology education and training of healthcare workers [10–12] alongside a lack of consistent access to reliable laboratory measurement of SCr [13,14]. In response to these concerns, and within the 0by25 framework, the ISN developed a protocol for AKD management in LLMICs. This included the development of a symptom-based risk score to screen for patients at increased AKD risk and the use of devices to measure SCr at the point-of-care (POC) [15]. These efforts were underpinned with AKD education programs delivered to healthcare workers providing care to patients presenting with acute illness at risk of AKD. The feasibility of this approach to early identification and management of AKD was tested in a pilot study that included 2,101 patients presenting at increased AKD risk to low-resourced regions in Malawi, Nepal, and Bolivia [15]. The protocol was shown to be feasible and effective, with AKD confirmed in 1,199 (57%) patients. Difficulties, however, were faced in following patients after healthcare facility discharge, with 36% of patients lost to follow-up within 1 month. Patient follow-up in LLMICs presents a specific challenge due to a lack of established electronic health records to track outpatient results alongside a frequent inability to measure serum creatinine outside of the hospital setting. This lack of healthcare infrastructure, alongside a deficiency in trained nephrology staff, low health literacy, and resource restrictions impacting patient ability to travel for healthcare visits, means opportunities for the early detection of CKD and its subsequent management are being missed.

Having proven the feasibility of this AKD management strategy, the ISN subsequently established the Kidney Care Network (KCN) [16]. The aim of this service improvement project is to implement the 0by25 AKD management approach into routine clinical care in low-resource settings. The project was undertaken in 4 LLMICs over a 2-year period and utilized an updated AKD risk score, as outlined in the section below. The effectiveness of this approach in identifying AKD and the outcomes in AKD after management with this protocol as part of routine clinical care is unknown.

The main objectives of this study were to investigate the feasibility and performance of an updated symptom-based risk score to screen for AKD in LLMICs when applied as part of

routine clinical care and to establish both patient and kidney outcomes from AKD in the short and medium term. We also aimed to provide further epidemiological data on AKD presenting in LLMICs including common causes and their management, in addition to the clinical variables associated with the development of AKD and its outcomes.

## Methods

### Ethics statement

Ethics approval was granted locally at each of the 4 study sites by the following ethics boards: the ethical committee of the Escola de Enfermagem da USP (University of Sao Paulo Nursing School), Brazil, approval 31670214; The Comité Regional de Enseñanza e Investigación, Hospital Obrero No 2—Caja Nacional de Salud, Cochabamba, Bolivia; the Nepal Health Research Council, Kathmandu, approval 205/2016; and the UKZN biomedical research ethics committee, South Africa, approval BE257/19. Consent was written in Brazil and Nepal and verbal in Bolivia. The requirement for consent was waived by the ethics board in South Africa as the project was categorized as a service improvement initiative.

### Study design, setting, and participants

The study was undertaken as part of the KCN project in low-resourced regions of Brazil, Bolivia, South Africa, and Nepal. Patients were recruited from a variety of healthcare facility types including healthcare centers (HCCs), district hospitals, and tertiary hospitals (**Table A in S1 Text**). A protocol to identify and manage AKD was instituted. In short, an education and training program were delivered to healthcare workers working at each of the study sites on the management of AKD. The training was site specific and included face-face workshops delivered over multiple days. These were run by local nephrologists and attended by multidisciplinary healthcare professionals (clinicians, nurses, physician assistants) providing clinical care at the sites of project implementation. A symptom-based risk score coupled with the provision of devices to measure SCr at the POC were used to facilitate AKD identification. The risk score was developed using data from the cohort of patients presenting in the previous 0by25 Pilot Feasibility Study [15]. A logistic regression analysis was undertaken to determine the clinical variables associated with AKD in this study (variable included in this analysis are outlined in **Table B in S1 Text**); points within the scoring system are attributed to symptoms associated with AKD (**Table 1**). The area under the receiver operating characteristic curve was 0.824 for the risk score to detect AKD with an optimal cut-off score of 10 points (sensitivity 92.9% and specificity 58.9% at this cut-off) based on data from the pilot study [15]. As such, a score of 10 points or more was considered to represent increased risk of AKD, and these patients underwent SCr testing. Patients with a risk score of <10 points at presentation could also be considered at risk of AKD and undergo SCr testing according to the judgment of the clinical team.

Adult (≥18 years) patients presenting to study sites between September 2018 and November 2020 were eligible to be screened for risk of AKD using the approach described. Those at increased kidney disease risk who underwent SCr testing were included. Patients on dialysis or with a kidney transplant, and those with missing data for presenting SCr or age category, were excluded. We report an observational cohort of patients managed with this approach. The size of the cohort reported represents a convenience sample of all patients managed within the pre-specified timeframe of the project; a sample size calculation was not performed. The management of patients with AKD was left to the discretion of the treating clinician, who also determined contributors to the development of AKD and the most likely primary cause. Patient and kidney outcomes were recorded at the end of the healthcare facility admission and

**Table 1. AKD risk score components.** A total score of ≥10 points represents increased risk of AKD.

| Factor | Points |
|---|---|
| Vomiting | 4 |
| Low oral intake | 2 |
| Weakness | 2 |
| Oliguria reported by patient | 8 |
| Hypotension | 8 |
| Appetite loss | 8 |
| Swelling | 5 |

Variable description

Vomiting–presence of dehydration associated with vomiting as determined by clinical team.

Low oral intake–presence of dehydration associated with low oral intake as determined by clinical team.

Weakness–reported by patient.

Oliguria–reported by patient.

Hypotension–blood pressure <90/60 mmHg or relative hypotension as determined by clinical team.

Loss of Appetite–acute or chronic symptom reported by patient.

Swelling–presence of non-traumatic swelling on limbs, face, or entire body.

at 90 days thereafter. Of note, in the manuscript we refer to healthcare "admission" which includes both patients who attended a healthcare facility, underwent SCr testing (+- the relevant management) and were discharged on the same day, in addition to those patients that were admitted for an inpatient stay.

## Variables and data sources/measurement

Clinical data were recorded prospectively at 3 time points: enrollment, at the end of the healthcare facility admission, and at post-discharge follow-up of up to 90 days. Devices to measure SCr at the POC were provided to each site (StatSensor Xpress CREA, Nova Biomedical, Waltham, Massachusetts, United States of America) [17,18]. SCr was either measured by the POC device or by an automated analyzer in a local laboratory. Clinical data were recorded on electronic devices using REDCap software (https://www.project-redcap.org) and exported as Microsoft Excel files for subsequent analysis.

## Definitions

Glomerular filtration rate was estimated (eGFR) using the CKD-EPI equation without race adjustment [19,20]. Kidney disease was defined according to KDIGO SCr functional criteria (https://kdigo.org) and classified as either AKD or CKD (**Table 2**). In accordance with the latest KDIGO consensus statement, AKD was defined by "abnormalities of kidney function and/ or structure with a duration of <3 months"; it was separated into AKD with and without AKI [21]. AKI was diagnosed and staged according to KDIGO criteria [22]. The latest SCr documented prior to healthcare facility admission and the lowest SCr during healthcare facility admission were used to determine the baseline SCr; an imputed baseline SCr based on an assumed eGFR was not used [23]. Urine output measurement and urinalysis data were not captured. Patients were categorized into those with and without kidney disease, and the nature of kidney disease was determined: AKD with AKI; AKD without AKI; or CKD. Kidney outcomes are defined in **Table 2.**

**Table 2. Definitions of kidney disease and kidney recovery.**

| KIDNEY DISEASE | | |
|---|---|---|
| **Kidney disease** | **Definition** | **Comments** |
| *AKD with AKI* | Increase in SCr by 50% within 7 days OR Increase in SCr by 0.3 mg/dl within 2 days from baseline | Stage 1: SCr increase by 1.5–1.9 times baseline; Stage 2: SCr increase by 2.0–2.9 times baseline; Stage 3: SCr increase by ≥3 times baseline or increase in SCr to ≥4 mg/dl or initiated on KRT Urine output criteria not used as data not captured |
| *AKD without AKI* | eGFR <60 ml/min/1.73 m$^2$ OR Decrease in eGFR by ≥35% OR Increase in SCr by 50% occurring over ≤3 months (but not within 7 days) | GFR estimated by CKD-EPI equation (2021) Structural criteria (urinalysis) not used as data not captured |
| *CKD* | eGFR <60 ml/min/1.73 m$^2$ for >3 months | |
| *NKD* | Not fulfilling criteria for AKD or CKD | |
| *Baseline SCr* | Latest creatinine documented prior to healthcare facility admission OR Lowest creatinine during healthcare facility admission | Lowest value of 2 criteria used Imputed baseline creatinine based on an assumed eGFR not used |
| KIDNEY RECOVERY | | |
| *Complete recovery* | Has follow up creatinine and last recorded creatinine has returned to within 0.1 mg/dl of baseline value AND Last recorded eGFR is ≥60 ml/min/1.73 m$^2$ | No ongoing kidney disease |
| *Partial recovery / persistent kidney disease* | Has follow up creatinine and last recorded creatinine is less than highest creatinine but remains >0.1 mg/dl above baseline OR Creatinine has recovered to baseline but last recorded eGFR is <60 ml/min/1.73 m$^2$ | |
| *No recovery / persistent kidney disease* | Has follow up creatinine and last recorded creatinine is highest creatinine during admission/follow-up period OR Remains dependent on KRT | Includes subset of patients on dialysis |
| *Unknown* | Creatinine not repeated during admission/ follow-up period | |

AKD, acute kidney disease; AKI, acute kidney injury; CKD, chronic kidney disease; NKD, no kidney disease; SCr, serum creatinine; eGFR, estimated glomerular filtration rate; KRT, kidney replacement therapy.

## Outcome measures

Primary outcome measures included the prevalence of AKD in the enrolled cohort and the performance of the risk score to detect it. In addition, patient mortality and kidney outcome at end of healthcare facility admission and at 90-day follow-up were determined. Secondary outcomes included the causes of AKD and the treatments used in its management. We also compared clinical variables (demographics [age and sex], healthcare facility type where patient enrolled, and AKD risk score at patient presentation) and outcomes (patient mortality and kidney outcomes as described in **Table 2**) between those with AKD and no kidney disease (NKD), and investigated variables associated with AKD development and mortality.

## Statistical methods

Data are presented as number and percentages for categorical variables and mean and standard deviation (SD) or median and interquartile range (IQR) for numerical variables depending on data distribution. Categorical variables were compared using the Fisher's exact or Chi-squared test. Numerical variables were compared between 2 groups using the Mann–Whitney or an unpaired *t* test. Variables are compared across greater than 2 groups with a one-way analysis of variance. Multivariable logistic regression analysis was undertaken to determine factors associated with the development of AKD and mortality. Age, sex, country of enrolment, and risk score at presentation were included in the model for AKD development; the same variables in addition to the presence of AKD were included in the model for mortality. Odds ratios (OR) and 95% confidence intervals (CIs) determined for each variable. Variables were selected as these data were prespecified as required in all participants at study enrollment and due to differences in these variables in patients with AKD and in patient survival in univariable analyses. The performance of the risk score was assessed using the area under the receiver operating characteristics curve and with a sensitivity and specificity analysis. The optimal score was determined by Youden's index. Youden's index is defined by sensitivity + specificity–1; it may be used to determine the cut-off representing the maximum potential effectiveness of the risk score. Analysis was performed using Graphpad Prism version 9 ([www.graphpad.com](www.graphpad.com)). A *p*-value of ≤0.05 was considered statistically significant. A formal prospective analysis plan was not used; analyses were determined after data collection. This study is reported as per the Strengthening the Reporting of Observational Studies in Epidemiology (STROBE) guideline (**S1 STROBE Checklist**).

## Results

### Participants

A total of 4,394 patients were screened for risk of AKD and 4,311 of these were deemed to be at increased AKD risk and enrolled (**Fig 1**), and 2,289 (53.1%) patients were female and median age was 57 (IQR 42–70) years. A total of 3,190 (74.0%) patients had an AKD risk score of ≥10 points, whereas 1,121 (26.0%) patients were deemed to be at risk of AKD by clinical judgment despite a risk score <10 points. Enrollment based on clinical judgment of AKD risk occurred predominantly at the South Africa site (**Table C in S1 Text**). Median risk score in all patients was 14 (IQR 8–19), ranging from 2 (IQR 0–8) in South Africa to 18 (IQR 14–22) in Nepal. The frequency of the presence of each component of the AKD risk score is outlined in **Table D in S1 Text**; reduced appetite and weakness were the most common symptoms. Data on the type of facility where patients presented were available in 4,293 patients; 1,356 (31.6%) patients presented to an HCC, 676 (15.7%) to a district hospital, and 2,259 (52.6%) to a tertiary hospital.

### Measurement of kidney function and prevalence of AKD

Creatinine was measured by POC device in 3,145 (73.0%) patients. Median enrollment creatinine and eGFR were 1.4 (IQR 1.0–1.9) mg/dl and 52 (IQR 34–78) ml/min/1.73m$^2$, respectively (**Fig A in S1 Text**). Enrollment eGFR was <60 ml/min/1.73 m$^2$ in 2,597 (60.2%) patients. A historical creatinine measured prior to enrollment was available in 1,239 (28.7%) patients.

Kidney disease was present in 2,959 (68.6%) patients, which included 2,922 (67.8%) patients with AKD and 37 (0.9%) patients with CKD (**Table 3 and Fig 1**), and 2,288 (53.1%) patients had AKD without AKI and 634 (14.7%) patients had AKD with AKI. Of the 634 patients with AKI, stage 1 was present in 391 (61.7%) patients, stage 2 in 140 (22.1%) patients, and stage 3 in

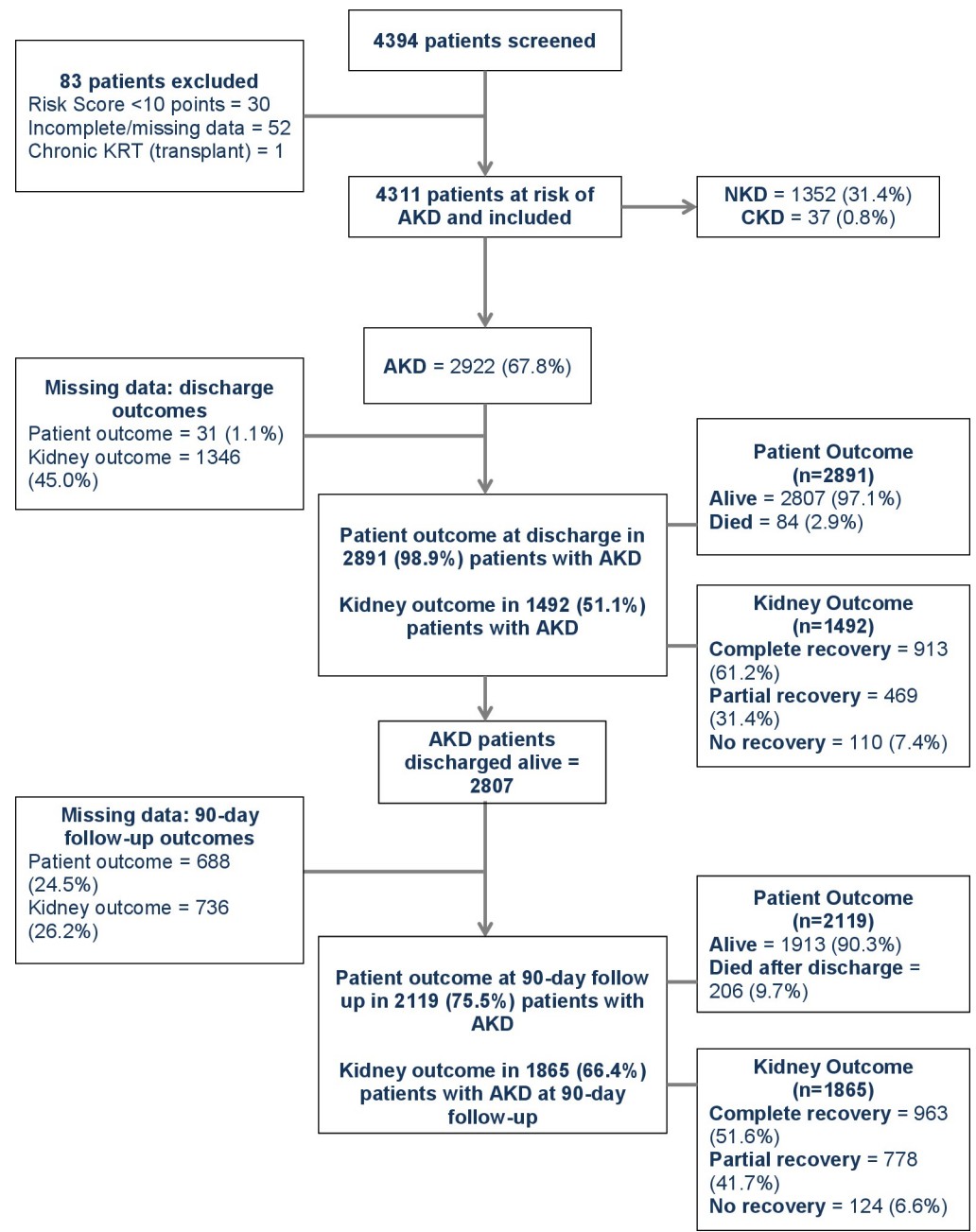

**Fig 1. Cohort description.** AKD, acute kidney disease; CKD, chronic kidney disease; KRT, kidney replacement therapy; NKD, no kidney disease.

103 (16.3%) patients. Patients with AKD were older than patients with NKD, a higher proportion were male, and they were more commonly enrolled from a tertiary hospital than an HCC (**Table 4**). In the multivariable analysis age (OR 1.04, 95% CI [1.03, 1.04]), risk score (OR 1.03, 95% CI [1.01, 1.04]), and presentation in Nepal (OR for presentation in Bolivia 0.38, 95% CI [0.31, 0.47], OR for presentation in Brazil 0.34, 95% CI [0.24, 0.49], OR for presentation in South Africa 0.09, 95% CI [0.07, 0.11]), were associated with the presence of AKD (**Table E in S1 Text**).

**Table 3. Measurement of kidney function and kidney disease classification at each study site.**

|  | Bolivia | Brazil | Nepal | South Africa | All patients |
|---|---|---|---|---|---|
| **Measurement of kidney function** | | | | | |
| Number with data | 951 | 197 | 1,952 | 1,211 | **4,311** |
| Enrollment SCr (mg/dL) (median; IQR) | 1.4 (1.0–1.9) | 1.2 (0.9–1.8) | 1.7 (1.4–2.3) | 1.0 (0.8–1.2) | **1.4 (1.0–1.9)** |
| Enrollment eGFR (ml/min/1.73 m$^2$) (median; IQR) | 51 (33–75) | 60 (36–85) | 41 (28–56) | 75 (57–93) | **52 (34–78)** |
| Enrollment eGFR <60 ml/min/1.73 m$^2$ (n; %) | 612 (64.4) | 97 (49.2) | 1,550 (79.4) | 338 (27.9) | **2,597 (60.2)** |
| SCr measured by POC device (n; %) | 431 (45.3) | 194 (98.5) | 1,309 (67.2) | 1,211 (100) | **3,145 (73.0)** |
| SCr documented prior to enrollment (n; %) | 325 (34.2) | 64 (32.5) | 609 (31.2) | 241 (19.9) | **1,239 (28.7)** |
| **Kidney disease classification** | | | | | |
| Number with data | 951 | 197 | 1,952 | 1,211 | **4,311** |
| AKD with AKI (n; %) | 116 (12.2) | 14 (7.1) | 404 (20.7) | 100 (8.3) | **634 (14.7)** |
| AKI Stage 1 (n; % of AKI) | 60 (51.7) | 7 (50.0) | 235 (58.2) | 89 (89.0) | **391 (61.7)** |
| AKI Stage 2 (n; % of AKI) | 39 (33.6) | 1 (7.1) | 93 (23.0) | 7 (7.0) | **140 (22.1)** |
| AKI Stage 3 (n; % of AKI) | 17 (14.7) | 6 (42.9) | 76 (18.8) | 4 (4.0) | **103 (16.3)** |
| AKD without AKI (n; %) | 568 (59.7) | 123 (62.4) | 1,295 (66.3) | 302 (24.9) | **2,288 (53.1)** |
| CKD (n; %) | 28 (2.9) | 4 (2.0) | 5 (0.3) | 0 (0.0) | **37 (0.9)** |
| NKD (n; %) | 239 (25.1) | 56 (28.4) | 248 (12.7) | 809 (66.8) | **1,352 (31.4)** |

AKD, acute kidney disease; AKI, acute kidney injury; CKD, chronic kidney disease; eGFR, estimated glomerular filtration rate; IQR, interquartile range; NKD, no kidney disease; POC, point of care; SCr, serum creatinine.

## Performance of the risk score to detect AKD

AKD was present in 2,557 (80.2%) of the 3,190 patients enrolled based on the risk score and 365 (32.5%) of the 1,121 patients enrolled according to clinical judgment ($p < 0.0001$). The median risk scores in patients with AKD and NKD were 15 (IQR 12–20) and 8 (IQR 0–15), respectively ($p < 0.0001$; **Table 4** and **Fig 2**). In this cohort, the area under the receiver

**Table 4. Clinical variables and patient outcomes in patients with AKD and NKD.**

|  | AKD | NKD | *p*-Value |
|---|---|---|---|
| **Demographic** | | | |
| Sex, Female (n; %) | 1,476 (50.5) | 792 (58.6) | <0.0001 |
| Age (years, median; IQR) | 61 (46–72) | 59 (43–70) | 0.0005 |
| **Healthcare facility type where patient enrolled** | | | |
| HCC | 458 (15.8) | 897 (66.8) | <0.0001 |
| District hospital | 520 (17.9) | 147 (11.0) | |
| Tertiary hospital | 1,928 (66.3) | 297 (22.1) | |
| Other/missing | 1 (0.0) | 1 (0.1) | |
| **Risk score** | | | |
| Total points (median; IQR) | 15 (12–20) | 8 (0–15) | <0.0001 |
| **Healthcare facility outcome** | | | |
| Number with data | 2891 | 1339 | |
| Died (n; %) | 84 (2.9) | 3 (0.2) | <0.0001 |
| **Outcome at 90-day follow up** | | | |
| Number with data | 2,119 | 422 | |
| Died (after discharge) | 206 (9.3) | 31 (7.3) | 0.14 |

AKD, acute kidney disease; HCC, healthcare centre; IQR, interquartile range; NKD, no kidney disease.

A. Risk score (individual values with median and IQR plotted) in patients with AKD and NKD

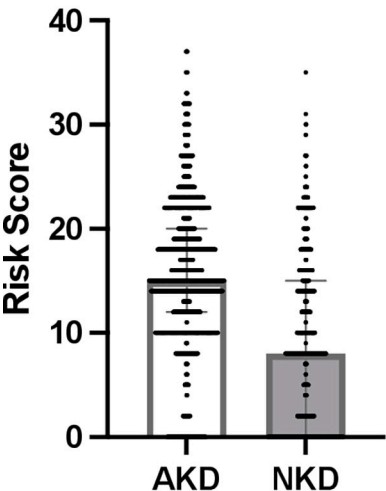

B. Receiver operating characteristic curve for the risk score to detect AKD

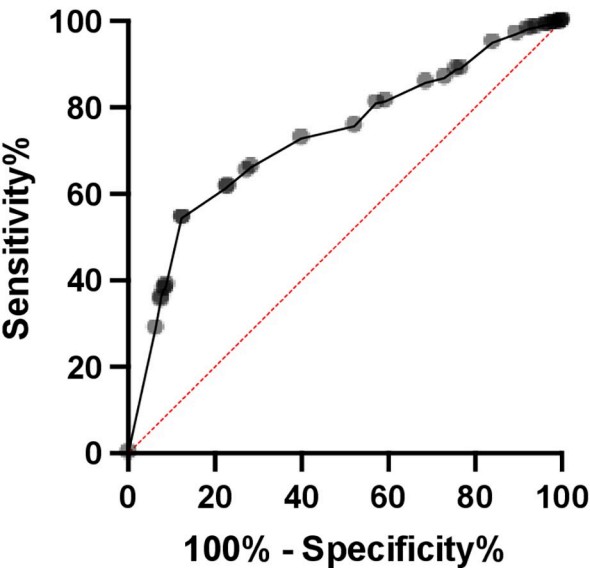

**Fig 2. Performance of the risk score to detect AKD.** The score attributes points to clinical features associated with AKD, with a higher total score representing increased AKD risk. (A) Risk score (individual values with median and IQR plotted) in patients with AKD and NKD. (B) Receiver operating characteristic curve for the risk score to detect AKD. AKD, acute kidney disease; IQR, interquartile range; NKD, no kidney disease.

operating characteristic curve was 0.73 for the risk score to detect AKD with an optimal cut-off score of 11 points ($p < 0.0001$; sensitivity 61.6% and specificity 77.2% at this cut-off; **Fig 2**).

## Causes and management of AKD

Contributors to the development of AKD, the primary causes of AKD, and strategies used in their management are outlined in **Table 5**. The main contributors to the development of AKD

**Table 5. Aetiologies and management of AKD.**

| | Bolivia | Brazil | Nepal | South Africa | All patients |
|---|---|---|---|---|---|
| **Contributors to the development of AKD*** | | | | | |
| Number with data | 681 | 132 | 1,696 | 402 | **2,911** |
| Dehydration (*n*; %) | 419 (61.5) | 19 (14.4) | 86 (5.1) | 33 (8.2) | **557 (19.1)** |
| Hypotension/shock (*n*; %) | 94 (13.8) | 5 (3.8) | 444 (26.2) | 18 (4.5) | **561 (19.3)** |
| Trauma (*n*; %) | 7 (1.0) | 0 (0.0) | 23 (1.4) | 5 (1.2) | **35 (1.2)** |
| Surgery (*n*; %) | 12 (1.8) | 0 (0.0) | 0 (0.0) | 3 (0.7) | **15 (0.5)** |
| Infection (*n*; %) | 420 (61.7) | 64 (48.5) | 757 (44.6) | 39 (9.7) | **1,280 (44.0)** |
| HIV (*n*; %) | 17 (2.5) | 0 (0.0) | 4 (0.2) | 23 (5.7) | **44 (1.5)** |
| Urinary obstruction (*n*; %) | 77 (11.3) | 9 (6.8) | 44 (2.6) | 9 (2.2) | **139 (4.8)** |
| Pregnancy related (*n*; %) | 13 (1.9) | 0 (0.0) | 7 (0.4) | 3 (0.7) | **23 (0.8)** |
| Allergic reaction (*n*; %) | 2 (0.3) | 0 (0.0) | 3 (0.2) | 1 (0.2) | **6 (0.2)** |
| Herbal medications (*n*; %) | 63 (9.3) | 0 (0.0) | 1 (0.1) | 2 (0.5) | **66 (2.3)** |
| Induced by other medications (*n*; %) | 139 (20.4) | 0 (0.0) | 20 (1.2) | 33 (8.2) | **192 (6.6)** |
| Poisoning (*n*; %) | 13 (1.9) | 1 (0.1) | 14 (0.8) | 3 (0.7) | **31 (1.1)** |
| Animal/insect bite (*n*; %) | 10 (1.5) | 18 (13.6) | 0 (0.0) | 1 (0.2) | **29 (1.0)** |
| Cardiorenal (*n*; %) | 27 (4.0) | 17 (12.9) | 178 (10.5) | 1 (0.2) | **223 (7.7)** |
| Hepatorenal (*n*; %) | 11 (1.6) | 1 (0.8) | 199 (11.7) | 1 (0.2 | **212 (7.3)** |
| Other (*n*; %) | 139 (20.4) | 49 (37.1) | 690 (40.7) | 171 (42.5) | **1,049 (36.0)** |
| **Primary cause of AKD*** | | | | | |
| Number with data | 665 | 132 | 1,695 | 284 | **2,776** |
| Dehydration (*n*; %) | 138 (20.8) | 13 (9.8) | 35 (2.1) | 24 (8.5) | **210 (7.6)** |
| Hypotension/shock (*n*; %) | 34 (5.1) | 5 (3.8) | 180 (10.6) | 11 (3.9) | **230 (8.3)** |
| Trauma (*n*; %) | 3 (0.5) | 0 (0.0) | 22 (1.3) | 4 (1.4) | **29 (1.0)** |
| Surgery (*n*; %) | 5 (0.8) | 0 (0.0) | 2 (0.1) | 2 (0.7) | **9 (0.3)** |
| Infection (*n*; %) | 301 (45.3) | 46 (34.8) | 753 (44.4) | 27 (9.5) | **1,127 (40.6)** |
| HIV (*n*; %) | 6 (0.9) | 0 (0.0) | 2 (0.1) | 14 (4.9) | **22 (0.8)** |
| Urinary obstruction (*n*; %) | 47 (7.1) | 7 (5.3) | 44 (2.6) | 3 (1.1) | **101 (3.6)** |
| Pregnancy related (*n*; %) | 7 (1.1) | 0 (0.0) | 5 (0.3) | 1 (0.4) | **13 (0.5)** |
| Allergic reaction (*n*; %) | 2 (0.3) | 0 (0.0) | 3 (0.2) | 0 (0.0) | **5 (0.2)** |
| Induced by other medications (*n*; %) | 43 (6.5) | 0 (0.0) | 16 (0.9) | 26 (9.2) | **85 (3.1)** |
| Poisoning (*n*; %) | 10 (1.5) | 1 (0.8) | 14 (0.8) | 2 (0.7) | **27 (1.0)** |
| Animal/insect bite (*n*; %) | 8 (1.2) | 17 (12.9) | 0 (0.0) | 1 (0.4) | **26 (0.9)** |
| Cardiorenal (*n*; %) | 9 (1.4) | 17 (12.9) | 165 (9.7) | 1 (0.4) | **192 (6.9)** |
| Hepatorenal (*n*; %) | 5 (0.8) | 1 (0.8) | 176 (10.4) | 0 (0.0) | **182 (6.6)** |
| Other (*n*; %) | 47 (7.1) | 25 (18.9) | 278 (16.4) | 168 (59.2) | **518 (18.7)** |
| **Management of AKD** | | | | | |
| Number with data | 681 | 132 | 1,696 | 402 | **2,911** |
| Oral fluid (*n*; %) | 321 (47.1) | 11 (8.3) | 31 (1.8) | 26 (6.5) | **389 (13.4)** |
| IV fluid (*n*; %) | 488 (71.7) | 36 (27.3) | 1,459 (86.0) | 30 (7.5) | **2,013 (69.2)** |
| Blood products (*n*; %) | 31 (4.6) | 1 (0.7) | 18 (1.1) | 4 (1.0) | **54 (1.9)** |
| Antibiotics (*n*; %) | 406 (59.6) | 61 (46.2) | 1,466 (86.4) | 40 (10.0) | **1,973 (67.8)** |
| HIV therapy (*n*; %) | 16 (2.3) | 0 (0.0) | 3 (0.2) | 18 (4.5) | **37 (1.3)** |
| Anti-venom therapy (*n*; %) | 9 (1.3) | 10 (7.6) | 1 (0.1) | 1 (0.2) | **21 (0.7)** |
| Relief of urinary tract obstruction (*n*; %) | 58 (8.5) | 8 (6.1) | 15 (0.9) | 7 (1.7) | **88 (3.0)** |
| Antihypertensives (*n*; %) | 48 (7.0) | 16 (12.1) | 32 (1.9) | 11 (2.7) | **107 (3.7)** |
| Vasopressors (*n*; %) | 22 (3.2) | 1 (0.8) | 12 (0.7) | 1 (0.2) | **36 (1.2)** |

(*Continued*)

**Table 5.** (Continued)

| | Bolivia | Brazil | Nepal | South Africa | All patients |
|---|---|---|---|---|---|
| Diuretics (*n*; %) | 67 (9.8) | 23 (17.4) | 182 (10.7) | 9 (2.2) | **281 (9.7)** |

*Contributors and causes of AKD were determined by the clinical judgment of the treating clinician.

AKD, acute kidney disease; HIV, human immunodeficiency virus; IV, intravenous.

were infection, hypotension/shock, and dehydration, present in 1,280 (44.0%), 561 (19.3%), and 557 (19.1%) cases of AKD, respectively. In those with infection, the commonest types of infection were "other bacterial" infections (*n* = 805; 63.1%) and gastroenteritis (*n* = 365; 28.6%); the main sites of infection were the urinary (*n* = 523; 41.9%) and gastrointestinal (*n* = 401; 32.1%) tract (**Table F in S1 Text**). Infection was also the most common primary cause of AKD, responsible for 1,127 (40.6%) cases. The most common treatments were fluid resuscitation (intravenous in 2,013 [69.2%] patients and oral in 389 [13.4%] patients) and anti-biotics (1,973 [67.8%] patients). KRT was indicated in 32 (1.1%) patients and provided in 26 (81.3%) patients. Hemodialysis was the KRT modality used in all cases.

## Kidney and patient outcomes at end of healthcare facility admission

A total of 4,266 patients, including 2,891 patients with AKD, had a healthcare facility patient outcome recorded, and 4,172 (97.8%) patients were discharged alive, and 94 (2.2%) patients died. Death during healthcare facility admission was more common in patients with AKD (*n* = 84; 2.9%) than in patients with NKD (*n* = 3; 0.2%) (*p* < 0.0001; **Table 4**). In multivariable analysis, age (OR 1.03, 95% CI [1.01, 1.04]), risk score at presentation (OR 1.14, 95% CI [1.10, 1.17]), presentation in a country other than Nepal (OR for presentation in Bolivia 29,83, 95% CI [13.95, 77.47], OR for presentation in Brazil 10.30, 95% CI [2.90, 35.01], OR for presentation in South Africa 14.62, 95% CI [4.44, 49.41]), and the presence of AKD (OR 2.48, 95% CI [1.27, 5.33]) were associated with death during admission (**Table G in S1 Text**).

Kidney outcomes in patients with AKD are outlined in **Table 6**; outcomes in the subset of patients with AKD with AKI are outlined in **Table H in S1 Text**. Kidney function was not

**Table 6. Patient and kidney outcomes at healthcare facility discharge and at 90-day follow-up in patients with AKD.**

| | Bolivia | Brazil | Nepal | South Africa | Total |
|---|---|---|---|---|---|
| **At healthcare facility discharge** | | | | | |
| Number with data | 684 | 137 | 1,699 | 402 | **2,922** |
| Unknown (no creatinine after enrolment) (*n*; %) | 91 (13.3) | 23 (16.8) | 944 (55.6) | 288 (71.6) | **1,346 (46.1)** |
| Partial recovery (*n*; %) | 242 (35.4) | 21 (15.3) | 192 (11.3) | 14 (3.5) | **469 (16.1)** |
| Complete recovery (*n*; %) | 244 (35.7) | 67 (48.9) | 517 (30.4) | 85 (21.1) | **913 (31.2)** |
| No kidney recovery (*n*; %) | 39 (5.7) | 22 (16.1) | 40 (2.4) | 9 (2.2) | **110 (3.8)** |
| Died (*n*; %) | 68 (9.9) | 4 (2.9) | 6 (0.4) | 6 (1.5) | **84 (2.9)** |
| **At 90-day follow-up** | | | | | |
| Number with follow-up | 490 | 89 | 1,418 | 122 | **2,119** |
| Death after discharge (*n*; %) | 1 (0.2) | 8 (9.0) | 197 (13.9) | 0 (0.0) | **206 (9.7)** |
| Death during admission or post discharge follow-up (*n*; %) | 69 (14.1) | 12 (13.5) | 203 (14.3) | 6 (4.9) | **290 (13.7)** |
| Number with creatinine at follow-up (*n*; %) | 487 | 63 | 1,219 | 96 | **1,865** |
| Partial recovery (*n*; %) | 154 (31.6) | 29 (46.0) | 561 (46.0) | 34 (35.4) | **778 (41.7)** |
| Complete recovery (*n*; %) | 309 (63.4) | 20 (31.7) | 584 (47.9) | 50 (52.1) | **963 (51.6)** |
| No kidney recovery (*n*; %) | 24 (4.9) | 14 (22.2) | 74 (6.1) | 12 (12.5) | **124 (6.6)** |

repeated prior to discharge in 1,346 (46.1%) AKD patients and as such kidney status was unknown; 1,492 (51.1%) patients with AKD had kidney status reassessed prior to discharge and 579 (38.8%) of these patients were discharged with known persistent kidney disease (**Fig 1**).

### Kidney and patient outcomes at 90-day follow-up

A total of 2,564 patients, including 2,119 patients with AKD, had follow-up after healthcare facility discharge; this occurred at a median of 91 (90 to 92) days after study enrollment. Death after discharge occurred in 237 (9.2%) patients. There was no difference in mortality after discharge between patients with AKD ($n$ = 206; 9.7%) and NKD ($n$ = 31; 7.3%) ($p$ = 0.14). Death at any time up to 90 days after enrollment occurred in 290 (13.7%) patients with AKD (**Table 4**). In multivariable analysis, age (OR 1.03, 95% CI [1.02, 1.03]) and total risk score (OR 1.07, 95% CI [1.05, 1.09]) were associated with death at any stage during follow-up; there was a negative association of death at any stage with presentation in South Africa (OR 0.12, 95% CI [0.05, 0.26]) and Bolivia (OR 0.68, 95% CI [0.50, 0.89]) (**Table G in S1 Text**).

A total of 1,865 (66.4% of those discharged) patients with AKD had reassessment of kidney function at 90-day follow-up (**Fig 1**). Kidney outcomes in these patients are outlined in **Table 6**, and 902 (48.4%) patients had persistent kidney disease, including 12 patients who remained on dialysis (representing 0.6% of AKD patients followed to this time point). In 740 (39.7%) patients, eGFR was persistently <60 ml/min/1.73 m$^2$ over at least 90 days representing de novo or previously undiagnosed CKD.

## Discussion

In this study, we investigated the effectiveness of an AKD management strategy to identify patients with AKD in LLMICs, and we determined patient and kidney outcomes from AKD when this management strategy was implemented as part of routine clinical care. We did this at multiple sites in 4 countries across 3 continents in patients presenting to a variety of health-care facility types. We included a large number of patients ($n$ = 2,922) with AKD. We demonstrated that the use of a symptom-based risk score, underpinned by an AKD education program, improved detection of AKD compared to clinical judgment alone. We provide data to support previous findings of more AKD in patients presenting to higher-level healthcare facilities from causes that were treatable by relatively simple means [15,24]. We demonstrated that with early identification and treatment, the requirements for KRT and in-hospital mortality were low. We highlighted that a large proportion of patients were discharged from the healthcare facility with either unknown kidney status or known persistent kidney disease. Furthermore, our unique data demonstrate that 1 in 10 patients with AKD die in the 90 days following discharge and that around one half of AKD patients will have persistent kidney disease at 3 months, many of whom are reclassified with de novo or previously undiagnosed CKD.

Large numbers of acutely unwell patients present to healthcare facilities in LLMICs each day, many of whom may be at risk of AKD. It is impractical, both logistically and financially, to undertake SCr testing in all patients, and as such efforts to risk stratify patients must be made. As part of the 0by25 initiative a symptom-based risk score was developed for this purpose and its use was previously shown to be feasible in a pilot study [15]. The risk score was subsequently updated for use in the current study: the number of variables within the score was reduced from 10 to 7 to simplify its use; moreover, its performance was improved using clinical variables associated with the development of AKD from real-world data in patients presenting to LLMICs. To the best of our knowledge, this study is the first to test the performance of the updated risk score; this was done with the score employed during routine clinical

care and formed one of our main study objectives. We have demonstrated its use to be feasible and effective, with an area under the ROC curve of 0.73 for the score to detect AKD. We demonstrated a higher prevalence of AKD in patients enrolled based on the risk score compared to clinical judgment alone (80% versus 33%) and this may explain the reduction in AKD prevalence at the South Africa site where clinical judgment was predominantly used to determine AKD risk. Our data demonstrate a cut-off score of 11 (as opposed to 10) is optimal for AKD identification. As would be expected, greater specificity was demonstrated at this higher cut-off, which may be important when rationalizing the limited resources available for SCr measurement in LLMICs. The risk score was not only predictive of the presence of AKD, but we also found it to predict patient survival, both in the short and medium term, adding weight to its utility in stratifying overall patient risk at the time of presentation.

Through this study, we also provide important further data on the epidemiology of AKD in LLMICs. The cohort in this study was older than adults included in the pilot feasibility project and comparable to some higher income AKI cohorts [25,26]. The cohort was preselected for those at risk of AKD, and the prevalence of AKD in this study (67.8%) was similar to that in the feasibility project (66%). A comparison of the key findings in this study alongside the other main studies from the 0by25 initiative is outlined in **Table I in S1 Text**. As with previous studies, AKD was most common in patients presenting to higher levels of healthcare facility, while the predominant causes were related to infection and hypovolemia and as such treatable by relatively simple interventions. The requirement for KRT was low in this study, indicated in only 1.1% cases of AKD, and when AKI was confirmed, it was most commonly mild, more reflective of higher-income settings. This may result from the early identification of kidney disease facilitating timely interventions targeting treatable causes as described.

The other key objective of this study was to determine both the short- and medium-term outcomes from AKD. Importantly, we assessed these outcomes in the setting of AKD being managed as part of routine clinical care. Patient outcome at discharge was recorded in most patients (98.9%) with AKD. While healthcare facility mortality was low, it was higher in patients with AKD than NKD supporting the known impact of AKD on patient outcomes [27]. A significant proportion of patients left the healthcare facility without reassessment of kidney function. This may reflect deficiencies in the management strategy or a natural high turnover of patients at study sites; data on facility length of stay were not recorded. While most patients who had kidney function reassessed had some improvement in kidney function, 38.8% left the healthcare facility with persistent kidney disease providing evidence to support the need to monitor AKD patients after discharge.

A unique aspect of this study is the follow-up data after healthcare facility discharge, and 2,119 cases of AKD had a patient outcome recorded at 90-day follow-up and 1,865 patients had kidney function reassessed at this time point. This reflects 75.5% and 66.4% of patients discharged post AKD, respectively and represents one of the largest cohorts of AKD patients followed-up in LLMICs to date. Mortality post discharge in this study was 9.7%, this being similar to the post discharge mortality (10.3%) at the same time point in the feasibility study. Notable is the higher post discharge mortality compared to inpatient mortality. The reasons for this are unclear (cause of death was not recorded) but warrants further study. Moreover, this consistent finding across more than 1 study adds further support for the need to follow AKD patients closely after the initial episode. Post discharge mortality was not different in patients with AKD compared to NKD, albeit only a small proportion of patients with NKD was followed up. The overall mortality in AKD patients in this study of 13.7% is, however, lower than other previous studies from low-resourced parts of the world [8,28,29].

Further evidence for the need to monitor patients with AKD after discharge comes from the kidney outcomes determined at 3 months. Kidney disease was persistent in around one

half of AKD patients and in 39.7% a new diagnosis of CKD was made. Given the lack of historical creatinine measurements we are unable to say whether these patients had de novo or previously undiagnosed CKD, but this finding highlights the close interconnection between AKD and CKD syndromes [30,31]. In this project, we used the more recent KDIGO concept of AKD which includes any disorder of kidney function present for less than 3 months, in contrast to the previous Acute Disease Quality Initiative (ADQI) concept of AKD being kidney disease that persists from days 7 to 90 after an episode of AKI [32]. The latter concept has been used in other studies previously [33–36]. Both concepts highlight the important continuum between acute and chronic kidney diseases, and reveal knowledge gaps in mechanisms that may improve outcomes post AKD episode [37]. Our data lend weight to the potential value of "post-AKD" clinics, which would provide a unique opportunity to test interventions that may enhance recovery from AKD and that would facilitate the early diagnosis of CKD. Early identification of CKD is of particular importance in low-resourced regions to allow interventions to be instituted to prevent progression to ESKD and the need for costly chronic KRT. The 0by25 initiative advocates following a "5R approach" to the management of AKD with the last of these phases representing rehabilitation [8]. This phase of the 5R approach is often neglected but the follow-up data in this project reinforce its importance as part of an overall strategy to tackle kidney disease in its entirety in disadvantaged populations worldwide.

We designed this pragmatic observational study to assess AKD identification and outcomes as part of routine healthcare. This was undertaken in a variety of healthcare facilities managed within different health systems across 3 continents. This led to marked heterogeneity in clinical practice and outcomes between sites (e.g., significantly less AKD in South Africa; lower inpatient mortality in Nepal) and results should be interpreted with this in mind. Further, selection bias likely occurred during cohort identification, given the high proportion (98%) of patients deemed to be at AKD risk, and there may have also been selection bias in those patients with AKD that were followed up. The performance of the risk score reported is based on this preselected cohort, and hence this may not be representative of its performance in a broader population. We deliberately collected only a minimum data set, recording data that would be documented as part of standard clinical practice, and we do not have data on healthcare facility length of stay. As such, we are also missing a comprehensive assessment of serial SCr measurements and we did not record data on urine output and urinalysis, which may have led to an underestimation of AKD prevalence. Moreover, as in previous studies in LLMICs, most patients did not have a recorded baseline creatinine, which may have led to the misclassification of kidney status in some. We did not use an imputed baseline creatinine based on an assumed "normal" eGFR as we felt it inappropriate to make such an assumption in a population at high risk of kidney disease. The high number of patients reclassified with CKD during follow-up supported this assumption. Furthermore, there is a lack of data to inform what a "normal" eGFR is and whether this is the same across the diverse populations studied. We used the CKD-EPI equation to estimate GFR but are aware that based on recent findings this may have underestimated true prevalence of kidney disease, specifically in African patients [38]. We did not measure kidney function in those that were not deemed to be at risk for AKD, which would have allowed a more comprehensive analysis of the risk score. We undertook follow-up at 90 days to facilitate the determination of CKD prevalence, but follow-up beyond this time point was not undertaken. Whilst follow-up occurred in a large proportion of patients, it was not undertaken in all and there were natural variations in clinical approaches and degree of follow-up between the study sites. Similarly, variation in clinical expertise and resources available at different sites may have impacted outcomes, and we were not able to control for these issues. The study was not designed to detect a statistically significant change in clinical outcomes for patients albeit we frame outcomes in this cohort with

those in previous similar studies in the sections above. We included patients from a range of healthcare facilities across 3 continents and as such our findings are generalizable to many low-resourced healthcare systems but not necessarily all.

In conclusion, we have demonstrated the use of a symptom-based risk score is feasible and effective in helping identify patients at risk of AKD during routine clinical care in LLMICs. We have provided unique outcome data in many patients at multiple LLMIC sites. We have demonstrated a high mortality rate (9.7%) in patients in the 3-month period after AKD admission and the persistence of kidney disease in around one half of patients. Our findings support the ongoing development of AKD management strategies for use in LLMICs, which should include resource for close patient follow-up after the presenting episode.

## Supporting information

**S1 STROBE Checklist. STROBE Statement—checklist of items that should be included in reports of observational studies.**
(DOCX)

**S1 Text. Table A.** Study sites. **Table B.** Clinical variables included in the logistic regression analysis used to create the risk score. **Table C.** Enrollment and demographic data at each study site. **Table D.** Presence of individual components of the AKD risk score in all patients and at each site. **Table E.** Multivariable analysis of factors associated with the development of AKD. **Table F.** Type of infection and main site of infection in patients with AKD in whom infection was a contributor to AKD development. **Table G.** Multivariable analysis of factors associated with mortality. **Table H.** Patient and kidney outcomes at healthcare facility discharge and at 90-day follow-up in patients with AKD with AKI. **Table I.** Prevalence, causes of kidney disease, and clinical outcomes in this and other studies within the 0by25 initiative. **Fig A.** Histograms of enrollment creatinine and eGFR in the entire cohort.
(DOCX)

## Acknowledgments

We thank the healthcare workers who provided clinical care during this study and, above all, the patients for their participation. We acknowledge the work of the ISN in obtaining the funding for the study and for their administrative support throughout. We thank Nova Biomedical (https://www.novabiomedical.com) for their in-kind support.

## Author Contributions

**Conceptualization:** Rhys D. R. Evans, Rolando Claure-Del Granado, Brett Cullis, Emmanuel A. Burdmann, David C. Harris, Mike V. Rocco.

**Data curation:** Rhys D. R. Evans, Martyn Fredlund, Mike V. Rocco.

**Formal analysis:** Rhys D. R. Evans.

**Funding acquisition:** Kelly Hendricks, Mike V. Rocco.

**Investigation:** Sanjib K. Sharma, Rolando Claure-Del Granado, Brett Cullis, Emmanuel A. Burdmann, FOS Franca, Junio Aguiar, Martyn Fredlund, Maria F. Iturricha-Caceres, Mamit Rai, Bhupendra Shah, Shyam Kafle.

**Project administration:** Rhys D. R. Evans, Rolando Claure-Del Granado, Brett Cullis, Emmanuel A. Burdmann, FOS Franca, Junio Aguiar, Martyn Fredlund, Kelly Hendricks, Maria F. Iturricha-Caceres, Mamit Rai, Bhupendra Shah, Shyam Kafle, Mike V. Rocco.

**Supervision:** Rhys D. R. Evans, Sanjib K. Sharma, Rolando Claure-Del Granado, Brett Cullis, Emmanuel A. Burdmann, Martyn Fredlund, David C. Harris, Mike V. Rocco.

**Validation:** David C. Harris, Mike V. Rocco.

**Visualization:** David C. Harris.

**Writing – original draft:** Rhys D. R. Evans, Mike V. Rocco.

**Writing – review & editing:** Rhys D. R. Evans, Sanjib K. Sharma, Rolando Claure-Del Granado, Brett Cullis, Emmanuel A. Burdmann, FOS Franca, Junio Aguiar, Martyn Fredlund, Kelly Hendricks, Maria F. Iturricha-Caceres, Mamit Rai, Bhupendra Shah, Shyam Kafle, David C. Harris, Mike V. Rocco.

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
