## [Editor Report · Decision Letter 0]

4 Jul 2024

Dear Dr Evans, 

Thank you for submitting your manuscript entitled "Identification and outcomes of acute kidney disease in patients presenting in low-resource settings" for consideration by PLOS Medicine.

Your manuscript has now been evaluated by the PLOS Medicine editorial staff and I am writing to let you know that we would like to send your submission out for external peer review.

Please re-submit your manuscript within two working days, i.e. by Jul 08 2024.

Feel free to email me at atosun@plos.org or us at plosmedicine@plos.org if you have any queries relating to your submission.

Kind regards,

Alexandra Tosun, PhD

Associate Editor

PLOS Medicine

---

## [Decision Letter · Decision Letter 1]

7 Aug 2024

Dear Dr Evans,

Many thanks for submitting your manuscript "Identification and outcomes of acute kidney disease in patients presenting in low-resource settings" (PMEDICINE-D-24-02124R1) to PLOS Medicine. The paper has been reviewed by subject experts and a statistician; their comments are included below and can also be accessed here: [LINK]

As you will see, the reviews were positive, but the reviewers, especially the statistical reviewer, pointed out a general lack of detail in the manuscript. After discussing the paper with the editorial team and an academic editor with relevant expertise, I'm pleased to invite you to revise the paper in response to the reviewers' comments. We plan to send the revised paper to some or all of the original reviewers, and we cannot provide any guarantees at this stage regarding publication.

We ask that you submit your revision by Aug 28 2024. However, if this deadline is not feasible, please contact me by email, and we can discuss a suitable alternative.

Don't hesitate to contact me directly with any questions (atosun@plos.org). 

Best regards, 

Alexandra 

Alexandra Tosun, PhD 

Associate Editor

PLOS Medicine

atosun@plos.org

Comments from the academic editor:

1. Interpretation of the data is limited by a problem of selection bias at every stage of the study. 98% of those screened were deemed to be at high risk for AKD, implying a high degree of selection. Follow-up after discharge occurred in only 59% and repeat creatinine testing occurred in 64% of those with AKD. In addition it is clear that there was a big difference in practice in the centre in South Africa versus the other centres. The impact of this problem of selection bias should be discussed in more detail in the limitations section.

2. The paper would benefit from more discussion of the concept of AKD and why it is particularly relevant in low resource settings.

3. The study is reported to have recruited persons “presenting” to healthcare centres but it is implied that all were admitted. The authors should clarify this and perhaps specify that all participants were admitted.

4. The ROC curve in this case was based only on data from a highly selected group of patients considered to be at high risk. Performance in this population would not necessarily be representative of performance in a broader population.

5. More detail should be presented on the multivariable analyses.

Comments from the reviewers: 

Reviewer #1: The study evaluates the performance of a risk score for Acute Kidney Disease (AKD) identification and outcomes in low- and low-middle-income countries (LLMICs). Conducted across healthcare facilities in Bolivia, Brazil, South Africa, and Nepal from September 2018 to November 2020, adult patients were screened using a symptom-based risk score and clinical judgment. Serum creatinine testing confirmed AKD in 2922 out of 4311 at-risk patients (67.8%). The risk score's area under the curve was 0.73, indicating moderate accuracy. AKD prevalence was higher when using the risk score compared to clinical judgment alone (80.2% vs. 32.5%). Mortality during admission was 2.9% for AKD patients, and 9.7% after discharge, with 48.4% of AKD patients showing persistent kidney disease at follow-up. The study underscores the risk score's utility in AKD identification and the need for diligent follow-up in LLMICs. 

Although the study was carefully designed and a lot of data were collected over the study period, I found the methods section, particularly the statistical methods section, to be lacking. Below are my specific comments.

1. Please add line numbers for easier reference. 

2. Introduction: The introduction mentions that kidney disease disproportionately affects disadvantaged populations in LLMICs but lacks specific statistics or data to quantify this burden. The authors could include concrete data or statistics on the prevalence and impact of kidney disease in LLMICs to strengthen the argument and provide a clearer picture of the problem's scale.

3. Introduction: The pilot study results are mentioned, but the challenges faced, such as the high rate of loss to follow-up, are not discussed in depth. Consider providing a more detailed analysis of the pilot study's challenges, including reasons for the high loss to follow-up rate and potential strategies to mitigate this issue in future implementations.

4. Introduction: The introduction briefly mentions the establishment of the KCN but lacks details on its implementation and scope. It could be helpful to include more information about the KCN's implementation, such as the specific interventions, the roles of participating healthcare facilities, and how the updated AKD risk score is integrated into routine clinical care.

5. Method, Study Design, Setting, and Participants: It would be beneficial to provide a detailed description of the regression used to predict the risk score. Specifically, what type of regression was used and outline the variables included in the regression models.

6. Outcome Measures: It would be helpful to categorize the outcomes into primary and secondary outcomes. Additionally, specify the clinical variables and outcomes compared between the two groups. A detailed description of these variables and outcomes would enhance clarity.

7. Statistical Methods: Provide more details about the exact multivariable analysis performed to determine the factors associated with AKD and mortality. Clarify if logistic regression was used and explain why certain variables (e.g., age, sex, country of enrollment) were included in the models. Describe the selection process for these variables. Additionally, explain Youden's index in the main text for better understanding. Overall, the statistical methods section needs more comprehensive information.

Reviewer #2: In this study, the authors attempted to validate the performance of the AKD risk stratification tool. According to reference 10, the initial design and rationale of this AKD tool are based on the pre-hospital kidney disease classification (NKD, AKD, CKD). This AKD concept might slightly differ from the AKD concept used in large cohort studies or from the concept according to the ADQI consensus (non-recovery status after AKI). However, considering prior exposure to a community-acquired AKI episode, this concept might still be applicable.

Comment 1: Should Table 2 be cited as Table 1 (with the current Table 1 changing to Table 2) according to the main text in the manuscript?

Comment 2: According to reference 18, the authors decided to use baseline creatinine based on "The latest SCr documented prior to healthcare facility admission and the lowest SCr during healthcare facility admission were used to determine the baseline SCr; an imputed baseline SCr based on an assumed eGFR was not used." Did the authors consider using the lowest creatinine level from -7 to -365 days before this index admission as baseline creatinine, as well?

Comment 3: I understand the authors followed the KDIGO AKD concept. However, I suggest that the authors also discuss the slightly different concept of AKD from ADQI. As mentioned above, the score used to identify patients at risk for AKD might account for an under-diagnosed community-acquired AKI episode prior. The authors should also cite the ADQI reference in this article.

ref 1: Acute kidney disease and renal recovery: consensus report of the Acute Disease Quality Initiative (ADQI) 16 Workgroup. Nat Rev Nephrol. 2017 Apr;13(4):241-257. doi: 10.1038/nrneph.2017.2

Comment 4: In the discussion section, I suggest the authors also discuss the post-AKD follow-up and the differences in patient outcomes in this study compared to other prior studies. Additionally, please cite some recently published articles provided below.

ref 1: Unveiling the enigma of acute kidney disease: predicting prognosis, exploring interventions, and embracing a multidisciplinary approach. Kidney Res Clin Pract. 2024 Jul;43(4):406-416. doi: 10.23876/j.krcp.23.289

ref 2: Incidence and Transition of Acute Kidney Injury, Acute Kidney Disease to Chronic Kidney Disease after Acute Type A Aortic Dissection Surgery. J Clin Med. 2021 Oct 18;10(20):4769.

ref 3: Comprehensive versus standard care in post-severe acute kidney injury survivors, a randomized controlled trial. Crit Care. 2021 Aug 31;25(1):322. doi: 10.1186/s13054-021-03747-7

ref 4: Acute Kidney Disease After Acute Decompensated Heart Failure. Kidney Int Rep. 2022 Jan 3;7(3):526-536. doi: 10.1016/j.ekir.2021.12.033

ref 4: Quality of Care for Acute Kidney Disease: Current Knowledge Gaps and Future Directions. Kidney Int Rep. 2020 Aug 6;5(10):1634-1642. doi: 10.1016/j.ekir.2020.07.031

Comment 5: For the causes of AKD listed in Table 5, could the authors provide supplemental table or appendix information regarding how each etiology for AKD, such as infection, cardiorenal, and hepatorenal, was defined?

Comment 6: Could the authors also provide the long-term kidney outcomes (such as the need for maintenance dialysis) or MAKE-90 after identifying AKD patients in this cohort? (since the authors mentioned "We undertook follow-up at 90 days")

Comment 7: The 90-day mortality after AKD is reported as "13.7%." This seems a little high. Do the authors have further discussion or explanation, with more detailed information? (since the in-hospital rate only 2.9% ?)

Comment 8: According to reference 10, the AKI episode after admission might be considered a recurrent AKI episode or AKD/AKI progression following a previously unidentified community AKI episode (as shown in reference 10, Fig 1: the AKI develops and is identified after AKD). Did the authors use the same concept in this article (AKD first, then AKI)? If so, please clarify this more clearly in the introduction and methods section since, as mentioned above, this differs from the ADQI AKD concept.

Reviewer #3: This is a very interesting and important work performed by Evans et al. The researchers included adult patients who were at increased risk of AKD and who had a creatinine measurement performed.

Few queries

1) Could you please clarify why the multivariable analysis includes the presence of AKD in the model if the study aims to compare clinical variables and outcomes between those with AKD and those without kidney disease (NKD)?

2) If the inclusion criteria was availability of creatinine measurement, I am unsure why 28.7% of patients had creatinine prior to enrolment - "Those at increased kidney disease risk who underwent SCr testing were included. Patients on dialysis or with a kidney transplant, and those with missing data for presenting SCr or age category, were excluded."

3) Can Table 3 show number of patients please.

4) I am unclear with regards to serum creatinine measurement. Did the author measure serum creatinine by POC at enrolment for patients who did not have pre-enrolment creatinine? Or, some patients had creatinine measure twice - pre-enrolment and at the time of enrolment?

5) Authors need to provide a brief on Education and training program delivered to healthcare workers.

6) The multivariable analysis has been included as supplementary table instead of being in the main document. 

Additional minor comment:

1) There are too many tables

---

* Please upload any figures associated with your paper as individual TIF or EPS files with 300dpi resolution at resubmission; please read our figure guidelines for more information on our requirements: http://journals.plos.org/plosmedicine/s/figures. While revising your submission, please upload your figure files to the PACE digital diagnostic tool, https://pacev2.apexcovantage.com/. PACE helps ensure that figures meet PLOS requirements. To use PACE, you must first register as a user. Then, login and navigate to the UPLOAD tab, where you will find detailed instructions on how to use the tool. If you encounter any issues or have any questions when using PACE, please email us at PLOSMedicine@plos.org.

* All authors must declare their relevant competing interests per the PLOS policy, which can be seen here:

https://journals.plos.org/plosmedicine/s/competing-interests

For authors with ties to industry, please indicate whether any of the interests has a financial stake in the results of the current study.

* PLOS Medicine requires that the de-identified data underlying the specific results in a published article be made available, without restrictions on access, in a public repository or as Supporting Information at the time of article publication, provided it is legal and ethical to do so. Please see the policy at http://journals.plos.org/plosmedicine/s/data-availability

and FAQs at http://journals.plos.org/plosmedicine/s/data-availability#loc-faqs-for-data-policy

The Data Availability Statement (DAS) requires revision. For each data source used in your study: 

* Please provide the name(s) of the institutional review board(s) that provided ethical approval. Please specify whether informed consent was written or oral.

FIGURES AND TABLES

SUPPLEMENTARY MATERIAL

REFERENCES

STUDY TYPE-SPECIFIC REQUESTS 

* Abstract: Please include the study design, population and setting, number of participants, years during which the study took place (enrollment and follow up), length of follow up, and main outcome measures.

* Please ensure that the study is reported according to the RECORD guideline (available from https://www.record-statement.org) and include the completed checklist as Supporting Information. Please add the following statement, or similar, to the Methods: "This study is reported as per the Reporting of Studies Conducted using Observational Routinely-Collected Data (RECORD) guideline (S1 Checklist)." When completing the checklist, please use section and paragraph numbers, rather than page numbers. Please note that we have also found the Standards for Reporting Implementation Studies (StaRI) statement and checklist that may be appropriate for your study. Please check and use as appropriate.

* For all observational studies, in the manuscript text, please indicate: (1) the specific hypotheses you intended to test, (2) the analytical methods by which you planned to test them, (3) the analyses you actually performed, and (4) when reported analyses differ from those that were planned, transparent explanations for differences that affect the reliability of the study's results. If a reported analysis was performed based on an interesting but unanticipated pattern in the data, please be clear that the analysis was data driven. 

* Please state in the Methods section whether the study had a prospective protocol or analysis plan. If a prospective analysis plan (from your funding proposal, IRB or other ethics committee submission, study protocol, or other planning document written before analyzing the data) was used in designing the study, please include the relevant document(s) with your revised manuscript as a Supporting Information file to be published alongside your study and cite it in the Methods section. A legend for this file should be included at the end of your manuscript. If no such document exists, please make sure that the Methods section transparently describes when analyses were planned, and when/why any data-driven changes to analyses took place. Changes in the analysis, including those made in response to peer review comments, should be identified as such in the Methods section of the paper, with rationale.

---

## [Decision Letter · Decision Letter 2]

2 Oct 2024

Dear Dr. Evans,

Thank you very much for re-submitting your manuscript "Identification and outcomes of acute kidney disease in patients presenting in low-resource settings" (PMEDICINE-D-24-02124R2) for review by PLOS Medicine.

Thank you for your detailed response to the reviewers' comments. I have discussed the paper with my colleagues and the academic editor, and it has also been seen again by two of the original reviewers. The changes made to the paper were mostly satisfactory to the reviewers. As such, we intend to accept the paper for publication, pending your attention to the reviewers' and editors' comments below in a further revision. When submitting your revised paper, please once again include a detailed point-by-point response to the editorial comments.

[LINK]

In revising the manuscript for further consideration here, please ensure you address the specific points made by each reviewer and the editors. In your rebuttal letter you should indicate your response to the reviewers' and editors' comments and the changes you have made in the manuscript. Please submit a clean version of the paper as the main article file. A version with changes marked must also be uploaded as a marked up manuscript file. Please also check the guidelines for revised papers at http://journals.plos.org/plosmedicine/s/revising-your-manuscript for any that apply to your paper.

We ask that you submit your revision within 1 week (Oct 09 2024). However, if this deadline is not feasible, please contact me by email, and we can discuss a suitable alternative.

Please do not hesitate to contact me directly with any questions (atosun@plos.org). If you reply directly to this message, please be sure to 'Reply All' so your message comes directly to my inbox.

We look forward to receiving the revised manuscript.

Sincerely,

Alexandra Tosun, PhD

Associate Editor 

PLOS Medicine

plosmedicine.org

ACADEMIC EDITOR COMMENTS

The authors have responded comprehensively and adequately to the reviewers' comments. The study still has multiple limitations but importantly these are now more completely acknowledged and discussed. There are a few typo's to correct and I picked up two minor language issues as follows:

Line76-77: “…high mortality from the acute AKD episode…” suggest rephrase to “…high mortality from the acute episode…” or “…high mortality from the AKD episode…”

Line 93-94: “This challenge is a consequence, at least in part, due to a deficiency in nephrology education…” suggest rephrase to: “This challenge is a consequence, at least in part, of a deficiency in nephrology education…”

Requests from Editors:

Please note that we always require a point-by-point response to not only reviewer comments, but all editorial comments, including general editorial requests. Please be sure to provide such a document.

FINANCIAL DISCLOSURE

The funding statement should include: specific grant numbers, initials of authors who received each award, URLs to sponsors’ websites. Also, please state whether any sponsors or funders (other than the named authors) played any role in study design, data collection and analysis, the decision to publish, or preparation of the manuscript. If they had no role in the research, include this sentence: “The funders had no role in study design, data collection and analysis, decision to publish, or preparation of the manuscript.”

DATA AVAILABILITY 

The Data Availability Statement (DAS) requires revision. For each data source used in your study: 

CODE AVAILABILITY

We expect all researchers with submissions to PLOS in which author-generated code underpins the findings in the manuscript to make all author-generated code available without restrictions upon publication of the work. In cases where code is central to the manuscript, we may require the code to be made available as a condition of publication. Authors are responsible for ensuring that the code is reusable and well documented. Please make any custom code available, either as part of your data deposition or as a supplementary file. Please add a sentence to your data availability statement regarding any code used in the study, e.g. "The code used in the analysis is available from Github [URL] and archived in Zenodo [DOI link]" Please review our guidelines at https://journals.plos.org/plosmedicine/s/materials-software-and-code-sharing and ensure that your code is shared in a way that follows best practice and facilitates reproducibility and reuse. Because Github depositions can be readily changed or deleted, we encourage you to make a permanent DOI'd copy (e.g. in Zenodo) and provide the URL.

TITLE

We suggest changing the title to “Identification and outcomes of acute kidney disease in patients presenting in Bolivia, Brazil, South Africa and Nepal”.

ABSTRACT

1) Abstract: Please structure your abstract using the PLOS Medicine headings (Background, Methods and Findings, Conclusions). Please combine the Methods and Findings sections into one section.

2) l. 36: Please define ‘AKD’ at first use.

3) In the last sentence of the Abstract Methods and Findings section, please describe the main limitation(s) of the study's methodology.

AUTHOR SUMMARY

We ask that you include a short, non-technical Author Summary of your research to make findings accessible to a wide audience that includes both scientists and non-scientists. The Author Summary should immediately follow the Abstract in your revised manuscript. This text is subject to editorial change and should be distinct from the scientific abstract. Ideally each sub-heading should contain 2-3 single sentence, concise bullet points containing the most salient points from your study. In the final bullet point of 'What Do These Findings Mean?', please include the main limitations of the study in non-technical language. Please see our author guidelines for more information: https://journals.plos.org/plosmedicine/s/revising-your-manuscript#loc-author-summary.

INTRODUCTION

1) Please cite the reference numbers in square brackets. Citations should precede punctuation. Please revise throughout the entire manuscript.

2) Please remove any subheadings from the Introduction section.

METHODS AND RESULTS

1) Please state in the Methods section whether the study had a prospective protocol or analysis plan. If a prospective analysis plan (from your funding proposal, IRB or other ethics committee submission, study protocol, or other planning document written before analyzing the data) was used in designing the study, please include the relevant document(s) with your revised manuscript as a Supporting Information file to be published alongside your study and cite it in the Methods section. A legend for this file should be included at the end of your manuscript. If no such document exists, please make sure that the Methods section transparently describes when analyses were planned, and when/why any data-driven changes to analyses took place. Changes in the analysis, including those made in response to peer review comments, should be identified as such in the Methods section of the paper, with rationale.

2) Please ensure that the study is reported according to the STROBE guideline (please use RECORD if you feel it is more appropriate), and include the completed STROBE checklist as Supporting Information. Please add the following statement, or similar, to the Methods: "This study is reported as per the Strengthening the Reporting of Observational Studies in Epidemiology (STROBE) guideline (S1 Checklist)."

3) l.247: We suggest introducing the abbreviation ‘CI’ for ‘confidence intervals’ here.

4) l.265, please state the statistical meaning of the numbers in parentheses, e.g. "57 (interquartile range (IQR): 42-70) years". Please revise throughout.

5) l.290ff: Throughout, we suggest reporting statistical information as follows to improve clarity for the reader "22% (95% CI [13%,28%]; p</=)". When reporting 95% CIs please separate upper and lower bounds with commas instead of hyphens as the latter can be confused with reporting of negative values. For example: “(OR 1.04, 95% CI [1.03-1.04])”

6) Table 1: Please spell out ‘AKD’ in the title or define ‘AKD’ below the table.

7) Table 3: Please define ‘POC’ and ‘IQR’ below the table.

8) Table 4: Please spell out ‘AKD’ and ‘NKD’ in the title or define both below the table. Please add a unit for age (years). Please define ‘HCC’, ‘IQR’. Please change the first demographic to “Sex, Female (n; %)”.

9) Table 5: Please spell out ‘AKD’ in the title or define ‘AKD’ below the table. Please define ‘HIV’ and ‘IV’. Please provide a definition of the numerical values (e.g. n (%)).

10) Table 6: Please spell out ‘AKD’ in the title or define ‘AKD’ below the table. Please provide a definition of the numerical values (e.g. n (%)).

11) Table 7: Please define ‘AKD’, ‘AKI’, ‘N/A’, ‘CKD’, ‘LLMIC’ and ‘HIC’.

12) Please provide titles and legends for all figures (including those in Supporting Information files).

13) Figure 1: Please define ‘AKD’, ‘KRT’, ‘NKD’, ‘CKD’.

14) Figure 2: Please define ‘AKD’ and ‘NKD’. Please indicate in the figure caption the meaning of the bars and whiskers as well as the lines and dots. Please add a brief explanation on the risk score.

DISCUSSION

1) Please remove any subheadings from the Discussion section.

2) l.396: Please temper claims of primacy of results by stating, "to our knowledge" or something similar.

3) Please note that we prefer not to introduce new tables/figures in the Discussion section. We suggest moving Table 7 to the Supplementary Material.

REFERENCES

1) PLOS uses the numbered citation (citation-sequence) method and first six authors, et al.

2) Please ensure that journal name abbreviations match those found in the National Center for Biotechnology Information (NCBI) databases (http://www.ncbi.nlm.nih.gov/nlmcatalog/journals), and are appropriately formatted and capitalised.

3) Where website addresses are cited, please include the complete URL and specify the date of access (e.g. [accessed: 12/06/2024]).

4) Please also see https://journals.plos.org/plosmedicine/s/submission-guidelines#loc-references for further details on reference formatting.

SUPPLEMENTARY MATERIAL

1) Please note that supplementary material will be posted as supplied by the authors. Therefore, please amend it according to the relevant comments outlined here and in the previous decision letter.

2) In the published article, supporting information files are accessed only through a hyperlink attached to the captions. For this reason, you must list captions at the end of your manuscript file. You may include a caption within the supporting information file itself, as long as that caption is also provided in the manuscript file. Do not submit a separate caption file.

When SI files are contained with a single file:

Please label the file as ‘S1 Supporting Information’.

Please apply alphabetical labelling to each table and figure contained within the S1 file. For example, ‘Fig A’ to ‘Fig Z’ and ‘Table A’ to ‘Table Z’.

Plain text does not need to be labelled and can just be given a title as necessary. For example, ‘Statistical Analysis Plan’.

Please cite tables/figures as ‘Fig A in S1 Supporting Information’ and/or ‘Table A in S1 Supporting Information’, for example.

Please cite plain text as, ‘Statistical Analysis Plan in S1 Supporting Information’, for example.

When SI files are uploaded as separate files:

Please label tables as ‘S1 Table’ (so on) and figures as ‘S1 Fig’ (and so on).

Any additional documents (protocols/analysis plans etc.) can be labelled as ‘S1 Protocol’, for example. Please cite items as exactly as labelled.

SOCIAL MEDIA

To help us extend the reach of your research, please provide any X (formerly known as Twitter) handle(s) that would be appropriate to tag, including your own, your co-authors’, your institution, funder, or lab. Please enter in the submission form any handles you wish to be included when we post about this paper.

Comments from Reviewers:

Reviewer #1: I would like to thank the authors for their thoughtful revisions in response to my and the other reviewers' comments. After reviewing the authors' responses and the revised manuscript, I am satisfied that they have adequately addressed the concerns raised. I am pleased to recommend this paper for acceptance and publication.

Reviewer #2: The authors have addressed my previous question regarding the different AKD definitions compared to the ADQI consensus and the recent KDIGO criteria. Additionally, detailed information about MAKE-90 and the etiologies of AKD/AKI has been provided in the revised manuscript.

[LINK]

General Editorial Requests

---

## [Editor Report · Decision Letter 3]

14 Oct 2024

Dear Dr. Evans,

Thank you very much for re-submitting your manuscript "Identification and outcomes of acute kidney disease in patients presenting in Bolivia, Brazil, South Africa and Nepal" (PMEDICINE-D-24-02124R3) to PLOS Medicine.

There are a few minor editorial issues that need to be addressed before we can accept the manuscript for publication; these are outlined at the end of this email. Please revise the paper accordingly, and submit the final revision within 1 week. 

A reminder that when your manuscript is accepted, an uncorrected proof of your manuscript will be published online ahead of the final version, unless you've already opted out via the online submission form. If, for any reason, you do not want an earlier version of your manuscript published online or are unsure if you have already indicated as such, please let the journal staff know immediately at plosmedicine@plos.org.

If you have any questions in the meantime, please contact me directly at hvanepps@plos.org. 

We look forward to receiving the revised manuscript by Oct 21st.   

Sincerely,

Heather Van Epps, PhD

Executive Editor, PLOS Medicine

[on behalf of]

Alexandra Tosun, PhD

Senior Editor 

PLOS Medicine

plosmedicine.org

Requests from Editors:

1. Abstract, ll 55-58: Please clarify the denominators used to calculate the percentages for persistent kidney disease (48.4%) and reclassification (39.7%). Should the denominator be the 2119 patients with AKD who had follow up (per line 368 in the Results)? If so, the percentage with persistent kidney disease would be 42.6% (902 of 2119), and the percentage who were reclassified would be 34.9% (740 of 2119). Please check and clarify if needed, and please accept my apologies if I have missed something. 

2. Please format your Author summary using bullet points, rather than a continuous narrative. Each sub-heading should contain 2-3 single sentence, concise bullet points containing the most salient points from your study. In the final bullet point of ‘What Do These Findings Mean?’, please include the main limitations of the study in non-technical language. Please see our author guidelines for more information: https://journals.plos.org/plosmedicine/s/revising-your-manuscript#loc-author-summary.

3. Results, ll 294-297; Please report the denominator used to calculate these percentages and explain why this is different from the total number (n=4311). Is this due to missing data for type of center visited?

4. Financial disclosure statement: please include grant numbers and the initials of the author to whom the grants were awarded. 

5. STROBE checklist – please remove page numbers (leaving only line numbers) as these will not correspond to the final published document.

---

## [Editor Report · Decision Letter 4]

22 Oct 2024

Dear Dr Evans, 

On behalf of my colleagues and the Academic Editor, Maarten W. Taal, I am pleased to inform you that we have agreed to publish your manuscript "Identification and outcomes of acute kidney disease in patients presenting in Bolivia, Brazil, South Africa and Nepal" (PMEDICINE-D-24-02124R4) in PLOS Medicine.

I appreciate your thorough responses to the reviewers' and editors' comments throughout the editorial process. We look forward to publishing your manuscript, and editorially there are only a few remaining minor stylistic/presentation points that should be addressed prior to publication. We will carefully check whether the changes have been made. If you have any questions or concerns regarding these final requests, please feel free to contact me at atosun@plos.org.

Please see below the minor points that we request you respond to:

1) Author Summary: Please introduce abbreviations the first time they are used (AKD, LLMICs) or spell them out if they are used only once, such as CKD.

2) STROBE checklist: We apologize for the miscommunication regarding page and line numbers. We ask that you replace the line numbers with paragraph numbers per section (e.g. "Methods, paragraph 1").

3) Citations: Please note and revise that multiple citations should be combined into a single pair of parentheses. For example: “Kidney disease disproportionately affects disadvantaged populations in low- and low-middle income countries (LLMICs) with poor access to care [1,2]”. Please note the lack of spacing between citations.

4) References: Where website addresses are cited, please use the word “accessed” when specifying the date of access (e.g. [accessed: 12/06/2024]).

5) In reference [15], please change ‘PLOS Med’ to ‘PLoS Med’.

Before your manuscript can be formally accepted you will need to complete some formatting changes, which you will receive in a follow up email (including the editorial points above). Please be aware that it may take several days for you to receive this email; during this time no action is required by you. Once you have received these formatting requests, please note that your manuscript will not be scheduled for publication until you have made the required changes.

PRESS

Sincerely, 

Alexandra Tosun, PhD 

Associate Editor 

PLOS Medicine